# Multiomics and Systems Biology Are Needed to Unravel the Complex Origins of Chronic Disease

**David Martino [1],[\*]** **, Rym Ben-Othman [2], Danny Harbeson [2] and Anthony Bosco [1],[\*]**

[1] Telethon Kids Institute; University of Western Australia, Nedlands, WA 6009, Australia
[2] Department of Pediatrics, Division of Infectious Diseases, University of British Columbia, Vancouver, BC V5Z 1M9, Canada; rbenothman@bcchr.ubc.ca (R.B.-O.); dannyharbeson@gmail.com (D.H.)
[\*] Correspondence: David.Martino@telethonkids.org.au (D.M.); Anthony.Bosco@telethonkids.org.au (A.B.)

**Abstract:** Modernization has now been linked to poor developmental experience, the onset of immune dysregulation and rising rates of chronic diseases in many parts of the world. Research across the epidemiological, clinical, and basic science domains supports the concept that poor developmental experience, particularly during prenatal life, can increase the risk of chronic disease, with enduring effects on long-term health. Single 'omics' approaches are ill-suited to dealing with the level of complexity that underpins immune dysregulation in early life. A more comprehensive systems-level view is afforded by combining multiple 'omics' datasets in order to delineate correlations across multiple resolutions of the genome, and of the genomes of the microorganisms that inhabit us. In this concept paper, we discuss multiomic approaches to studying immune dysregulation and highlight some of the challenges and opportunities afforded by this new domain of medical science.

**Keywords:** multiomics; dysregulation; immune system; development; modernity

---

## 1. Introduction

Over the past two centuries, environmental changes associated with the transition to modernity have brought about major demographic, epidemiological, and ecological changes which have had profound effects on human health [1]. Improvements in public health, hygiene, food security, transport, and communication, plus increasing population growth, declining global fertility rates, global migration, and increased exposure to industrial pollutants have had major effects on human immune development and the microbiota. On the scale of human evolution, spanning hundreds of thousands of years, these changes virtually happened yesterday morning, and are now associated with rising rates of chronic non-communicable diseases (NCDs) across the globe, due to mismatches between our evolved capacities and modern environments [2]. NCDs now account for the major causes of human mortality across the globe, whereas previously infectious disease constituted the major burden of human mortality [3].

Immune development is a target for the adverse effects of modernization since chronic inflammation and immune dysregulation are common risk factors for a range of NCDs, including allergy and autoimmune disease, atherosclerotic disease, diabetes, and cancer, and emerging evidence now suggests links to neurological disease (reviewed in [4]). Efforts to understand the mechanistic basis for immune dysregulation and NCDs suggest complex multifactorial pathways of disease induction. Given the multiple pathways through which immune trajectories can be manipulated by the sociodemographic changes associated with modernity, there is a need to embrace data-driven approaches of increasing complexity to better enable precision medicine. Traditional reductionist approaches are ill-suited to dealing with the emergent properties of complex biological systems, and even single 'omics' medicine is proving ill-equipped to meet these challenges [5]. In this article,

we discuss the potential opportunities afforded by integrated multiomic approaches powered by pattern-finding algorithms to better understand the basis for immune dysregulation over a life course.

## 2. Perinatal Influences on Immune Development and Chronic Disease

It is now recognized that the first thousand days of life between conception and age two are a crucial period of heightened plasticity, in which the developing brain and the immune system are highly sensitive to influences from the environment [6]. Poor developmental experience brought about by adverse exposures (e.g., maternal stress, poor nutrition, infections, toxins and pollutants, alcohol, tobacco smoke, and antibiotics) during the perinatal period have a greater capacity to alter developmental trajectories, potentially increasing the risk for chronic diseases [7]. Preterm birth is a prototypical example of a suboptimal developmental experience with well-described heath consequences in later life. Recent findings from a multiomic study of preterm neonates supports the notion of altered developmental trajectories. In this study, cellular immune profiles, blood transcriptomes, plasma proteins, and microbiome profiles were assessed at birth in cord blood, and at 1, 2, and 12 weeks of age in a cohort of 100 infants, delivered preterm or at term. These data, in combination with systems biology methods, demonstrated that preterm and term infants followed largely distinct developmental trajectories that converged by around three months of age. This convergence toward a stereotypical immune trajectory was ostensibly linked to microbial interactions in early life [8].

Microbial interactions have long been recognized as potent signals that promote immune development, and disruption of the microbial ecology brought by modernization is increasingly recognized as a key risk factor in chronic disease research [9,10]. Given immune development is intimately coupled to microbial signaling, dysbiosis of the microbiome in early childhood has been linked to virtually every chronic disease of the modern era [11]. Nowhere has this been more clearly demonstrated than in the Amish and Hutterite story. The Amish and Hutterite populations living in the United States have a similar genetic background, diet, and lifestyle, with the exception that the Amish use traditional farming methods and live in close proximity to their animals. By contrast, the Hutterites employ highly industrialized farming methods, and the animals are housed in large facilities away from their homes. Airborne dust samples collected from Amish homes were found to have elevated levels of microbial endotoxin, which was associated with markers of immune maturation from sampled blood, and a four-fold reduction in the rates of childhood asthma and allergic disorders [12]. Accordingly, other studies have also reported that disruptions to the neonatal microbiome is linked to diabetes, asthma, necrotizing enterocolitis, inflammatory bowel disease, obesity, and various other inflammatory diseases, as recently reviewed by Amenyogbe et al. [13].

In addition to the microbiome, other exposures that also constitute poor developmental experience operative during a sensitive window of early development are likely to affect developmental plasticity, causing the induction of disease risk with long-term consequences for health [14,15]. Changes in nutrition and lifestyles related to modernization have dramatically altered parental factors like age at conception, body mass at conception, metabolism, and stress with support from animal models and human studies, indicating these factors induce developmental programming of disease risk [16]. Epigenetic mechanisms have been proposed as key mediators of the effects of adverse developmental experience (reviewed in [16]). Critically, at the earliest stages of reproduction, the few cells forming the conceptus are fully exposed to conditions that disturb epigenetic mechanisms, leading to persistent alterations in embryonic gene expression that are heritable across cell divisions and alter developmental trajectories [15]. These mechanisms have been proposed as exponents of rapidly rising community rates of NCDs independently of genetic effects.

## 3. Moving beyond Reductionist Biology: Systems-Level Understanding of Immune Dysregulation

Elucidation of the mechanisms underlying perinatal induction of immune dysregulation will be extremely challenging because biological phenotypes are emergent properties of highly complex

and dynamic interactions between a large number of molecular components [17]. The advent of high-throughput sequencing and mass spectrometry now enables the profiling of these molecular components across multiple layers of regulation (genome, epigenome, transcriptome, proteome, metabolome, and microbiome). These omic technologies have proven invaluable for enhancing our understanding of the individual layers of regulation, however, efforts are underway to combine these individual layers into a more comprehensive multiomic view of the entire system. Parallel initiatives are underway to develop the tools to more comprehensively measure the exposome, which encompasses the totality of the environmental exposures encountered by an individual over their life course [18,19]. We anticipate that incorporation of individual omic data into multiomic space in combination with the exposome will enable a greater understanding of complex biological systems. These systems biology approaches will be necessary in order to make sense of the complexity of interactions that govern immune development and dysregulation.

Systems biology begins with the understanding that cellular behavior and function are emergent properties of complex and dynamic interactions between genes and biomolecules. Emergent properties are characteristics of the whole system that are not present in the individual molecular components and, hence, complex systems are more than the sum of their parts. The application of network graph theory to the study of complex systems has revealed that biological systems are governed by a set of universal organizing principles, or common design elements that can also be observed in man-made systems, such as power grids and the world wide web [20]. Complex biological systems have a *scale-free* architecture (Figure 1), where most genes are interlinked with a few genes, and a few genes interact with many, giving rise to hubs [21–23].

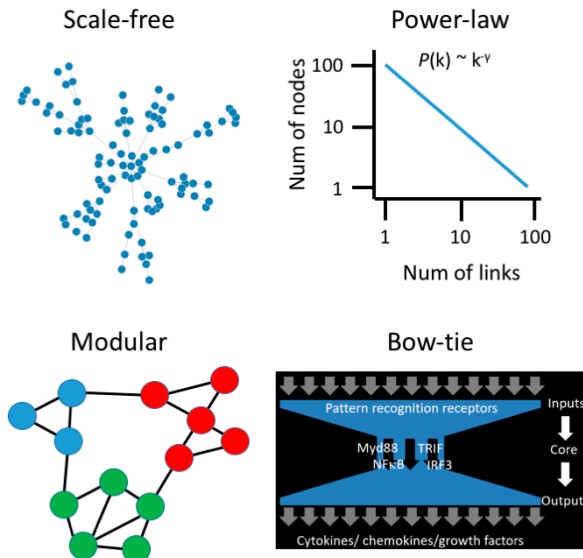

**Figure 1.** Emergent properties of biological networks. Scale-free networks follow a power-law degree distribution, meaning that most nodes have few links, and a few nodes have many links. Modules are subnets comprised of proteins that function in the same pathway. The innate immune system employs a bow-tie architecture to integrate diverse signals from a fluctuating environment.

As biological networks evolve and add new components, the probability of adding new links to an existing component is proportional to the number of links it already has, and this generates hubs through a mechanism known as *Preferential Attachment* or *Rich-get-Richer* [24]. Biological networks are also modular, meaning that genes which function in the same biological process form communities in the network structure (Figure 1). Smaller modules are embedded within larger modules, giving rise to *hierarchical organization*. Biological networks are *small world*, meaning that you can "hop" through the network from one gene to any other gene in a few steps. Finally, a key challenge for the immune system is to interpret and respond to diverse and fluctuating signals in order to maintain homeostasis.

This is achieved through a bow-tie signal processing control system [25]. The bow-tie structure receives diverse input signals and processes them through a core, and this generates complex output signals (Figure 1). The bow-tie architecture is an important conceptual advance in our understanding of immune dysregulation because genetic or environmental perturbation of the input and output layers will generate lots of noise and complexity but, ultimately, a limited number of core pathways control everything [26,27].

## 4. Utilizing Systems Biology Approaches for Very Early Prediction and Intervention for Immune-Mediated Diseases

Very early identification of high-risk children before they develop chronic diseases is extremely challenging because of the multitude of contributory genetic, environmental, and lifestyle factors. Recent foundational multiomic studies have begun to pioneer the approaches to identify disease and transitions and intervene early. In a landmark study, Price et al. followed 108 healthy adults for nine months, and collected biological samples (saliva, blood, urine, stool) every three months [26]. Multiomic profiles were generated, which included whole genome sequencing, 16S rRNA gut microbiome sequencing, 218 clinical diagnostic tests, 262 proteins, and 643 metabolites. The multiomic assays were designed to assess five health domains (cardiovascular, diabetes, inflammation, nutrition and toxins, stress). The genome sequencing data was summarized into polygenic risk scores for 127 disease traits. These risk scores are a single variable that estimates the genome-wide risk for a given trait by summing the number of risk alleles for each individual, weighted by effect size estimates from large genome-wide association studies (GWAS) [27]. The data were integrated by constructing an interomic correlation network, which captures pairwise interrelationships between the five omic layers. The $\alpha$-diversity (species richness) of the gut microbiome was positively correlated with height and $\beta$-nerve growth factor levels, and negatively correlated with levels of CSF-1, IL-8, and FLT3 ligand. Clinical diagnostic tests identified deviations from wellness or test results outside of normal reference ranges [28]. These insights were then leveraged to suggest evidence-based changes to diet (including supplements) and lifestyle (exercise, stress management) that resulted in significant improvements to biomarker levels across multiple health domains—type 2 diabetes (fasting glucose, HbA1c levels, insulin), cardiovascular disease (total cholesterol, LDL cholesterol), inflammation (IL-8, TNF), and toxins (mercury).

Chronic obstructive pulmonary disorder (COPD) is a highly heterogeneous, chronic, inflammatory lung disease which has early life origins in a subset of patients [29]. In a proof-of-concept study, Li et al. evaluated the utility of multiomic data to differentiate between COPD patients, healthy non-smokers, and smokers with normal lung function [5]. They found that the mean accuracy of subgroup prediction (healthy, smoker, COPD) was extremely poor when each of the omic data blocks were analyzed in isolation. However, combining data from multiple omic platforms increased the mean prediction accuracy to 100%, even when group sizes were limited to small numbers. These analyses highlight the potential for multiomic approaches to dramatically improve our understanding of highly complex and heterogeneous inflammatory diseases.

Very early identification of at-risk individuals from birth is now theoretically possible with polygenic risk scores [30]. These risk scores combine information derived from variants across the entire genome [30], and are able to identify segments of the population which are at heightened risk (more than three-fold) for a range of complex traits including inflammatory diseases [31]. One caveat of polygenic risk scores is that information is collapsed across the genome without taking into account the cellular or biological context. To address this issue, cluster analysis of deep immunological and clinical data can be utilized to stratify subjects into distinct developmental trajectories [32], and the polygenic risk scores can be leveraged to find those clusters enriched for subjects at heightened genetic risk. Extending this approach to multiomic layers will undoubtedly increase the resolution of these analyses and further refine the critical windows of opportunity for early intervention.

## 5. Multiomic Studies: Challenges and Opportunities

### 5.1. Sample Collection

Multiomic studies have been successfully applied to adults to identify molecular correlates of disease risk [26,33]. The application of these tools to early infancy is itself challenging because only limited volumes of blood can be collected, and collection of fasting blood is not feasible. Despite this, protocols for small volume collections yielding high-quality data have been developed and published. A recent study of 58 immune cell populations and 267 plasma protein immunoassays was performed on 100 μL of whole blood [8]. Whole blood represents a convenient and attractive tissue for multiomic studies that is amenable to fractionation into different aliquots for individual omic platforms that can be integrated following data generation. The caveat of whole blood is that it is a complex tissue, and omic profiles therefore represent an average profile across all cells in the blood sample, which can limit analyses of cell specific effects. Despite this, whole blood is likely to be the tissue of choice for multiomic studies with protocols in development that allow immune profiling, genetics, transcriptomics, epigenetics, metabolomics, and exposome profiling from less than 5 mL of blood.

In order to ensure data veracity, it is crucial to stringently standardize the processes of sample preparation even before considering the different omic assays. This strategy starts with sample processing, protocol testing, optimization, and establishment of gold standard procedures and, importantly, technological and biological controls. This focus on quality control includes assurance of lot-consistent performance of supplies and reagents allowing minimal deviations to ensure pristine data collection, compilation, assessment and, finally, analysis (Figure 2). Small differences in materials and supplies or deviations in procedure across sites can entirely derail a multiomic study.

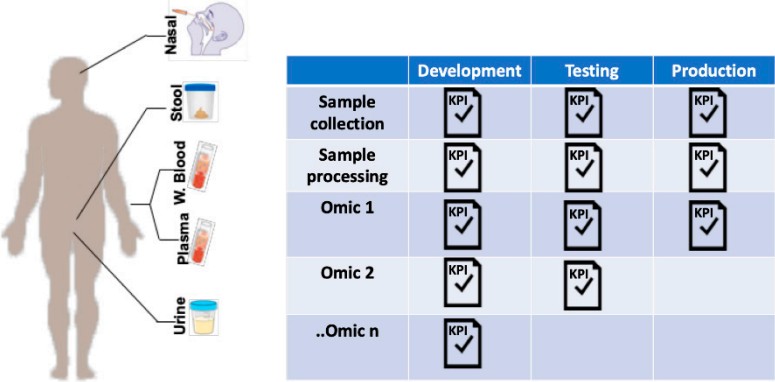

**Figure 2.** Production workflow for sample collection in multiomic studies. Standardization of sample collection can be systematically undertaken using a production workflow. For each sample type, protocols are developed and tested according to pre-determined key performance indicators (KPIs) before moving into production. KPIs are determined by investigators but could include standard quality control metrics published in the medical literature. The process is iterative, so the methodology is optimized if KPIs are not achieved, and proceed through re-testing until targets are met.

### 5.2. Data Collection

Researchers should identify data which may be important to downstream quality control. For instance, tracking the information about storage conditions of a sample, or the time it takes for the sample collection, not only informs about its quality but also can help discriminate outliers from subjects showing meaningful biological differences [34]. It is also very important to record the right metadata about the study subjects as this will be one of the major components of the multiomic data analysis. Correlations between multiomic data and clinical metadata often make up a major portion of the analysis, which is why it is extremely important to put careful consideration into what information will be collected prior to enrollment of the participants [35]. These clinical metadata are study-specific

and depend on the nature of the disease or the biological question asked. The identification of the metadata types can be challenging when looking at the impact of sociodemographic changes and modernity on immune development dysregulation. For instance, a study of immune development of neonates should consider prenatal and parental variations in environments, nutrition, lifestyle, age at conception, delivery mode, and many other factors. All of these could be a driving force behind different immune signatures revealed by multiomic analysis. By combining data types, we should be able to see the full spectrum of the global impact of the environment changes on immune development in an unprecedented manner, by leveraging the power of data integration [36].

*5.3. Data Management*

Appropriately managing the generated information, collecting the appropriate metrics, centralizing the analysis pipelines, and standardizing the entire process are key fundamental steps that are not easy to implement in large multiomic studies. A first line challenge in any multiomic study involves devising an appropriate strategy to store, index, organize, audit, distribute, and archive vast amounts of big data. Initiatives such as the open-source integrated Rule Orientated Data Systems (iRODS) provide an overall framework for data management tasks and allow complete management of primary and derived data with rules and policies to ensure reproducibility of the research [37].

This includes attaching clinical metadata and metrics, covering the entire range of steps from the bio-sample collection to processing of omic data files. Without a pre-defined strategy to build a common infrastructure for data storage, accessibility, and analysis (e.g., code reviewing), the data integrity backing the biological insight can quickly be jeopardized. In fact, the massive amount of data generated needs to be first verified for quality control purposes, and cleaned prior to being analyzed in a transparent and reproducible manner amongst the multiomic study experts. A second challenge involves the integration of clinical patient data with high-dimensional omic datasets. Software that enables integration of curated phenotypic data from clinical observations with biomarker data from gene expression and genotyping studies are emerging [38,39]. In order for multiomic studies to be reproducible, these computing and software infrastructures will be mission-critical over the long term.

*5.4. Data Analysis*

Bioinformatic approaches to extract meaning from multiomic data is an obvious challenge. Most of the unsupervised data integration methods require cutting edge tools and models using extensive computing capabilities to develop new algorithms and theoretical methods that fit the different layers of omic datasets [40,41]. The challenge for data integration is to apply the appropriate bioinformatic approaches to the study type and data collected. Moreover, the different omic datasets have a distinct format, size, and dimensionality, which represent one of the major computational challenges when it comes to data integration [42]. Advances in the field of data integration have made it possible to start generating guidelines on data integration approaches depending on the size of the dataset and its heterogeneity [43]. These methods and approaches are still in their infancy and, at present, highly specialized, but are anticipated to become more established as more researchers enter the multiomic space.

## 6. Concluding Remarks

Post-industrial changes to the environment are associated with rising rates of chronic non-communicable diseases in many parts of the world. Changes in the microbiome and the effect on immune development are firmly implicated, given the development of the microbiome in early life is particularly sensitive to even minor disturbances. Given the complexity and multifactorial nature of immune development, systems-level approaches are now needed to delineate trajectories of immune ontogeny. We posit, therefore, that integrative omics is expected to become increasingly influential for disease prediction, diagnosis, prevention, and prognosis. While these approaches are still in their infancy, the future development of multiomic workflows and community standards will

be necessary milestones toward capitalizing on the potential benefits of multilevel data integration. At present, the only cases of omic technologies that have translated to the clinic include genome sequencing and, to a lesser extent, RNA sequencing. Substantial regulatory and technical hurdles exist before other omic techniques will be approved for clinical use. Therefore, in the short-to-medium term, we anticipate integrative omics and associated models of disease risk will enhance the research enterprise and enable a clearer picture of health and disease. The initial stages toward uptake will need to initially demonstrate novel actionable insights and then prove through rigorous trial-based testing that early interventions designed from multiomic data do indeed provide tangible clinical benefits. In parallel, the data standards and precision around individual omic platforms will need to comply with levels acceptable for clinical tests in order to transition into the healthcare setting. Although the challenges are substantial, the potential benefits of multiomic studies necessitate research and development.

**Author Contributions:** Writing—Original Draft Preparation, D.M., R.B.O., D.H., A.B. Writing—Review & Editing, D.M., R.B.O., D.H., A.B.

**Funding:** This research received no external funding.

**Conflicts of Interest:** The authors declare no conflict of interest.

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
