# Peer review of "Multiomics and Systems Biology Are Needed to Unravel the Complex Origins of Chronic Disease"

_challenges, doi:10.3390/challe10010023_

Round 1

Reviewer 1 Report

The abstract is vague, mixed several ideas without congruence. 

There is only one reference for the entire first paragraph. Sentence from line 34-36 is lack of reference. Also this first paragraph looks a mixed of ideas without congruence and direction. 

In the second page, in lines 55-62 authors mentioned several ideas an a lot of information and only have one reference. That is correct? all this information can be bound in that reference?

Lines 76-79 authors wrote "other studies",implying that they are referencing information from several studies, but they get all the information from a review, so they do not cite those who obtained the primary information. Please reference de original sources.

Line 101-102, authors wrote "we anticipate that the incorporation...", which evidence the authors have to make that statement?

Line 201-207, it will be nice to outline that process in a figure.

Paragraph 209-225 is lack of any reference. all the ideas expressed in this paragraph are originals of the authors, or come from other articles?, On what evidence arte the authors bases to make these assertions?

Finally, I found lack of a clear conclusions. The authors provide us with a manuscript consisting of disconnected ideas and revolts, that do not have a clear direction in their speech and that in the end we do not reveal a clear conclusion or conclusions.

Author Response

Our kind thanks to the reviewer for providing feedback. Below we have addressed all the comments in the revised version of the manuscript.

R1: The abstract is vague, mixed several ideas without congruence. 

Response: We've modified the abstract for clarity so that it now reads:

Abstract: Modernization has now been linked to poor developmental experience, the onset of immune dysregulation and rising rates of chronic diseases in many parts of the world. Research across the epidemiological, clinical and basic science domains supports the concept that poor developmental experience particularly during prenatal life can increase the risk of chronic disease, with enduring effects on long-term health. Single ‘omics’ approaches are ill-suited to dealing with the level of complexity that underpins immune dysregulation in early life. A more comprehensive systems level view is afforded by combining multiple ‘omics’ data sets in order to delineate correlations across multiple resolutions of the genome, and that of the genomes of the microorganisms that inhabit us. In this concept paper we discuss multi-omics approaches to studying immune dysregulation and highlight some of the challenges and opportunities afforded by this new domain of medical science.

R1: There is only one reference for the entire first paragraph. Sentence from line 34-36 is lack of reference. Also this first paragraph looks a mixed of ideas without congruence and direction. 

Response: As this was a commentary piece rather than extensive review, the introduction contains a number of well accepted statements that were not referenced in the original paper or were views of our own. In the revision we have included additional supportive references, and edited for improved clarity. Sentence 34-36 does contain a reference [2] in the original manuscript so this was left unchanged.  The first paragraph now reads...

Over the past two centuries environmental changes associated with the transition to modernity have brought about major demographic, epidemiological and ecological changes which have had profound effects on human health [1]. Improvements in public health, hygiene, food security, transport and communication and increasing population growth, declining global fertility rates, global migration and increased exposure to industrial pollutants have had major effects on human immune development and the microbiota. On the scale of human evolution spanning hundreds of thousands of years these changes virtually happened yesterday morning, and are now associated with rising rates of chronic non-communicable diseases (NCDs) across the globe, due to mismatches between our evolved capacities and modern environments [2].  NCDs now account for the major causes of human mortality across the globe, whereas previously infectious disease constituted the major burden of human mortality [3].

R1: In the second page, in lines 55-62 authors mentioned several ideas an a lot of information and only have one reference. That is correct? all this information can be bound in that reference?

Response: Lines 55-62 discuss the findings and implications from one recent multiomic study of preterms. The reference to the original article is provided so this section is left unchanged.

R1: Lines 76-79 authors wrote "other studies",implying that they are referencing information from several studies, but they get all the information from a review, so they do not cite those who obtained the primary information. Please reference de original sources.

Response: We have have amended the text in the revision as follows:

"Accordingly, other studies have also reported that disruptions to the neonatal microbiome is linked to diabetes, asthma, necrotizing enterocolitis, inflammatory bowel disease, obesity and various other inflammatory diseases as recently reviewed by Amenyogbe et al [11]."

R1: Line 101-102, authors wrote "we anticipate that the incorporation...", which evidence the authors have to make that statement?

Response: This statement is an opinion of the authors.  

R1: Line 201-207, it will be nice to outline that process in a figure.

Response: We have now included a new figure 2 which depicts the sample collection optimisation in the form of a production workflow.

R1:Paragraph 209-225 is lack of any reference. all the ideas expressed in this paragraph are originals of the authors, or come from other articles?, On what evidence arte the authors bases to make these assertions?

Response: In the revision we have included additional supportive references. The paragraph now reads:

Researchers should identify data which may be important to downstream quality control. For instance, tracking the information about a sample’s storage condition or the time it takes for the sample collection, not only informs about its quality but also can help discriminate outliers from subjects showing meaningful biological differences [34]. It is also very important to record the right metadata about the study subjects as it will be one of the major components of the multi-omics data analysis. Correlations between multi-omics data and clinical metadata often make up a major portion of the analysis, which is why it is extremely important to put careful consideration into what information will be collected prior to the participants’ enrollment [35]. These clinical metadata are study specific and depend on the nature of the disease or the biological question asked. The identification of the metadata types can be challenging when looking at the impact of socio demographic changes and modernity on the immune development dysregulation. For instance, a study of immune development of neonates should consider prenatal and parental variations in environments, nutrition, lifestyle, age at conception, delivery mode, and many other factors. All of these could be a driving force behind different immune signatures revealed by multi-omic analysis. By combining data types, we should be able to see the full spectrum of the global impact of the environment changes on the immune development, in an unprecedented manner by leveraging the power of data integration [36].

R1:Finally, I found lack of a clear conclusions. The authors provide us with a manuscript consisting of disconnected ideas and revolts, that do not have a clear direction in their speech and that in the end we do not reveal a clear conclusion or conclusions.

Response: Thankyou we have revised the conclusions. We note that reviewer 2 commented the manuscript was coherent and well written, and we feel the conclusion does make several key points including 1) multiomic techniques are needed to understand immune development 2) workflows and community standards are only emerging in this area 3) there are substantial regulatory challenges for translation. Therefore we posit that substantial RnD is needed before the true benefits are realised, but the challenge is a worthy one. The revised conclusion now states:

Post-industrial changes to the environment are associated with rising rates of chronic non-communicable diseases in many parts of the world. Changes in the microbiome and the effect on immune development are firmly implicated, given the development of the microbiome in early life is particularly sensitive to even minor disturbances. Given the complexity and multi-factorial nature of immune development, systems-level approaches are now needed to delineate trajectories of immune ontogeny. We posit therefore that integrative omics is expected to become increasingly influential for disease prediction, diagnosis, prevention and prognosis. While these approaches are still in their infancy, the future development of multi-omic workflows and community standards will be necessary milestones toward capitalizing on the potential benefits of multi-level data integration. At present the only cases of omics technologies that have translated to clinic include genome sequencing and to a lesser extent, RNA sequencing. Substantial regulatory and technical hurdles exist before other omics techniques are approved for use in clinic. Therefore, in the short to medium term we anticipate integrative omics and associated models of disease risk will enhance the research enterprise and enable a clearer picture of health and disease. The initial stages toward uptake will need to initially demonstrate novel actionable insights, and then prove through rigorous trial-based testing that early interventions designed from multi-omic data do indeed provide tangible clinical benefits. In parallel the data standards and precision around individual omics platforms will need to comply with levels acceptable for clinical tests in order to transition into the healthcare setting. Although the challenges are substantial, the potential benefits of multi-omic studies necessitate research and development.   

Reviewer 2 Report

The manuscript titled "Multi-Omics and Systems Biology are Needed to Unravel the Complex Origins of Chronic Disease." Presents a perspective on the importance of using system biology techniques, specifically the combination of -omics approaches to understand chronic non-communicable diseases.

The paper is very well written, present a well structured and persuasive argument.  I believe the manuscript can be published without further delay.

I found one typo in the manuscript that should be fixed before publication.

- Page 3, line 126 "word" should be "world"

Author Response

Our kind thanks for the reviewer for the time taken to appraise the manuscript.

In the revision we have amended the spelling error as suggested.

With thanks

Reviewer 3 Report

The commentary attempts to shed light on the challenges and opportunities associated with employing multi-omics approaches to medical science. Considering the comprehensive summary of the current omics’ integration and systems biology approaches, this work is definitely interesting. However, there are a few comments that need to be addressed before this article can be recommended for publication.

1.    The section on “Perinatal influences on immune development and chronic disease” (Section 2) can see some improvement in terms of organization. Specifically, the last paragraph needs to be structured in a way to recapitulate the past studies and making constructive comments/remarks on the basis of that, instead of poorly coherent mentions of previous works.

2.    It is not clear why authors are discussing minute details in lines 112-115. While these lines are a concise description of the work(s) cited, the details do not serve any useful purpose in the flow of arguments in the paper. 

3.    The authors should check and correct the term “small world”. 

4.    The detailed description of the “bow-tie signal processing” in lines 127-139 is not necessary. The authors can summarize the published works in a shorter paragraph and focus more on moving ahead with systems-level understanding of immune dysregulation.

5.    The full form of “COPD” should be used when it first appears.

6.    The same comment as point 4 can be made for the lines 163-174. The authors are advised to shorten this section as well.

7.    The correct form of the part of speech should be used for the word “Specialize” in line 256.

Author Response

Our thanks goes to the reviewer for the time taken to appraise the manuscript. Below we have provided a point by point response. 

1.   The section on “Perinatal influences on immune development and chronic disease” (Section 2) can see some improvement in terms of organization. Specifically, the last paragraph needs to be structured in a way to recapitulate the past studies and making constructive comments/remarks on the basis of that, instead of poorly coherent mentions of previous works.

Response: In the revision we've amended the last paragraph for clarity as suggested. The new paragraph now reads:

In addition to the microbiome, other exposures that also constitute poor developmental experience operative during a sensitive window of early development are likely to affect developmental plasticity, causing the induction of disease risk, with long-term consequences for health [12,13]. Changes in nutrition and lifestyles related to modernization have dramatically altered parental factors like age at conception, body mass at conception, metabolism and stress with support from animal models and human studies that these factors induce developmental programming of disease risk (reviewed in [14]). Epigenetic mechanisms have been proposed as key mediators of the effects of adverse developmental experience [14]. Critically, at the earliest stages of reproduction when the few cells forming the conceptus are fully exposed to conditions that disturb epigenetic mechanisms, leading to persistent alterations in embryonic gene expression that are heritable across cell divisions and alter developmental trajectories [13]. These mechanisms have been proposed as exponents of rapidly rising community rates of NCDs independently of genetic effects.

2.    It is not clear why authors are discussing minute details in lines 112-115. While these lines are a concise description of the work(s) cited, the details do not serve any useful purpose in the flow of arguments in the paper. 

Response: We have shortened this paragraph for clarity. It now reads :

      Systems biology begins with the understanding that cellular behaviour and function are emergent properties of complex and dynamic interactions between genes and biomolecules. Emergent properties are characteristics of the whole system that are not present in the individual molecular components, and hence complex systems are more than the sum of their parts. The application of network graph theory to the study of complex systems has revealed that biological systems are governed by a set of universal organising principles, or common design elements that can also be observed in man-made systems, such as power grids, and the world-wide web [20]. Complex biological systems have a scale-free architecture, where most genes are interlinked with a few genes, and a few genes interact with many, giving rise to hubs [21] [22,23]

3.    The authors should check and correct the term “small world”. 

Response: Small world is a recognised property and accepted terminology of biological networks. 

4.    The detailed description of the “bow-tie signal processing” in lines 127-139 is not necessary. The authors can summarize the published works in a shorter paragraph and focus more on moving ahead with systems-level understanding of immune dysregulation.

Response: We have shortened this section for clarity. It now reads:

Finally, a key challenge for the immune system is to interpret and respond to diverse and fluctuating signals in order to maintain homeostasis. This is achieved through a bow-tie signal processing control system [25]. The bow-tie structure receives diverse input signals and processes them through a core, and this generates complex output signals. The bow-tie architecture is an important conceptual advance in our understanding of immune dysregulation, because genetic or environmental perturbation of the input and output layers will generate lots of noise and complexity, but ultimately a limited number of core pathways control everything [26,27].         

5.    The full form of “COPD” should be used when it first appears.

Response: Thankyou this has now been amended in the revision.

6.    The same comment as point 4 can be made for the lines 163-174. The authors are advised to shorten this section as well.

Response: we have now shortened for clarity. It now reads:

Chronic obstructive pulmonary disorder (COPD) is a highly heterogeneous, chronic, inflammatory lung disease, which has early life origins in a subset of patients [29]. In a proof-of-concept study, Li et al evaluated the utility of multi-omic data to differentiate between COPD patients, healthy non-smokers and smokers with normal lung function [5]. They found that the mean accuracy of subgroup prediction (Healthy, Smoker, COPD) was extremely poor when each of the omic data blocks were analyzed in isolation. However, combining data from multiple omic platforms increased the mean prediction accuracy to 100%, even when group sizes were limited to small numbers. These analyses highlight the potential for multi-omic approaches to dramatically improve our understanding of highly complex and heterogeneous inflammatory diseases.

7.    The correct form of the part of speech should be used for the word “Specialize” in line 256.

Response: this has now been amended

Round 2

Reviewer 1 Report

The manuscritp have improved a lot, good work.